# Post-Occupancy Evaluation in Post-Disaster Social Housing in a Hot-Humid Climate Zone in Mexico

Yarely Aguilar-Perez [1,*] , Lucelia Rodrigues [2,*] , Paolo Beccarelli [2] and Renata Tubelo [2]

1    Building, Energy and Environment Research Group, Faculty of Engineering, University of Nottingham, University Park, Nottingham NG7 2RD, UK
2    Department of Architecture & Built Environment, Faculty of Engineering, University of Nottingham, University Park, Nottingham NG7 2RD, UK; paolo.beccarelli@nottingham.ac.uk (P.B.); renata.tubelo@nottingham.ac.uk (R.T.)
*    Correspondence: yarely.aguilarperez@nottingham.ac.uk or arq.yap@gmail.com (Y.A.-P.); lucelia.rodrigues@nottingham.ac.uk (L.R.); Tel.: +44-0115-95-13176 (Y.A.-P.)

**Abstract:** In Mexico, the national fund for disasters (FONDEN) spent up to USD 800 million yearly building low-income housing for victims of highly destructive disasters. Since 2002, a total of 34 thousand new houses have been built as a response to Hurricane Isidore. However, recent research suggests that most of the FONDEN houses have been either abandoned or repurposed, which questions their suitability. In this paper, the authors sought to gain insight into occupants' perception of post-disaster social housing in Yucatan, southern Mexico, with views of understanding their use across this housing typology. The method employed in this study was a post-occupancy evaluation (POE), investigating occupants' satisfaction levels and thermal comfort in the homes. This was conducted through questionnaires, observation through photographic documentation, and environmental monitoring. Results revealed that 52% of occupants reported extreme dissatisfaction during warm seasons and 28% experienced dissatisfaction during cold seasons. The most used electrical appliance (84%) was found to be fans for increasing air movement. This was consistent with the results of the environmental monitoring, which demonstrated that there was thermal discomfort 67% of the time in September and 19% in December. This accounts for a warm-humid period and cool period, respectively. The results suggested that thermal discomfort may contribute to the abandonment and repurposing of these houses.

**Keywords:** post-disaster social housing; post-occupancy evaluation; thermal comfort; user satisfaction; hot-humid climate

## 1. Introduction

Disasters represent one of the main causes of damage to the built environment [1]. According to Sarkodie (2019) [2], developed countries and developing countries have different approaches to addressing risk assessment and preparedness frameworks, resulting in different levels of vulnerability to climatological hazards. According to Hallegatte et al. (2018) [3], developed countries typically rely on private insurance services to recover after a disaster, while developing countries, such as Mexico, rely on the government and non-governmental organisations (NGOs) to recover after a disaster.

In 2015, the UN defined 17 sustainable development goals (SDGs) to achieve by 2030 [4]. Goal number 11 is dedicated to making cities and human settlements inclusive, safe, resilient, and sustainable. One of the main four topics included in this goal is disaster risk reduction (DRR). This goal refers to the Sendai Framework [5] as the most up-to-date guidance on disaster response. Historically, a series of frameworks were created over time to respond to potential disasters, and these are the state-of-the-art in disaster recovery. At an international level, in 1994, the Yokohama Strategy and Plan of Action for a Safer World [6] was the first framework to set the links between disaster risk reduction and sustainability.

This was followed by the Kyoto Protocol in 1997 [7], Build Back Better (BBB) (in 2005) [8], and subsequently the Sendai Framework for disaster risk reduction (2015–2030) [9]. Within the scope of the Sendai Framework, key strategies are to enhance disaster preparedness for effective response and to "Build Back Better" in recovery, rehabilitation, and reconstruction. This means to recover not only to the original state, but to improve the quality and resilience of the society affected [9].

The BBB approach is embedded in the Sendai Framework, and it evolved over time. In 2019, Noy et al. [10] considered the three initial subcategories of BBB (stronger, faster, more inclusively) insufficient to describe the goals for recovery after a disaster. Four categories were then recommended and integrated into the scheme: (i) Build Back Safe, (ii) Build Back Fast, (iii) Build Back Fair, and (iv) Build Back Potential. The concept of safer rather than stronger was proposed because the concept of safety would include the surrounding context as well as the structural soundness of the built environment. For instance, considering wider roads and more green areas could diminish the wind strength in the built environment. The recovery speed is a clear goal for the BBB framework. However, it was acknowledged that a faster recovery could compromise community participation, engagement in decision making, and strategic planning. The Build Back Fair goal calls for inclusivity in the recovery efforts. The authors highlighted that low-income societies are commonly left behind and are a vulnerable population that might need extra effort to recover after a disaster. Building back with a potential focus on the need to enhance the economic opportunities and dynamism to recover after the disaster would help to make BBB sustainable for the victims.

Frameworks such as BBB set extensive guidance on recovery, but still lack a more in-depth analysis of how to deliver an effective recovery. A holistic framework for recovery should include economic, natural, social, and built environment considerations [11]. However, regardless of the existence of BBB in recovery management, very limited research had been done to quantify and evaluate the outcome effectiveness beyond assessing the speed at which the communities returned to their original state [12].

Specifically, housing is one of the most important and complex aspects of recovery after a disaster, since it represents a crucial element for comfort to the victims during the chaotic recovery phases after a disaster and is a fundamental condition for recovery [13]. Literature shows that post-disaster social housing worldwide suffers from several major drawbacks such as high rates of abandonment and repurposing [14]; low satisfaction rates among beneficiaries might be the leading cause of this [15]. However, few field investigations have paid attention to this issue [16–18].

Some authors have identified key factors that significatively influence satisfaction rates in post-disaster social housing, including inaccurate adaptation to local contexts, such as climate and cultural values [19], poor housing quality, design disregarding occupants' needs and expectations [20], thermal comfort [14,21], failure to fit their rural lifestyle [16,22], and limited financial and technical capacities to maintain modern houses [23]. Therefore, it becomes more urgent to improve post-disaster housing design to achieve resilience and sustainable development in vulnerable societies [14,16–18,20–22].

Within the scope of housing provision, FONDEN was a Mexican trust that oversaw post-disaster reconstruction (PDR) of public infrastructure and low-income homes from 1996 to 2020 [24]. FONDEN invested 800 million USD yearly in reconstruction throughout Mexico ([24] p.6). Nearly half of its resources were spent on the Yucatan Peninsula in 2002 through the reconstruction of homes, which is the focus of this investigation [25]. The FONDEN reconstruction programme aimed to deliver resilient housing in line with the BBB framework by offering structurally sound modern houses to the beneficiaries [24]. However, the adaptation aspect to the social and economic context seems to have not been met, delivering housing that might result in inadequate thermal comfort to the occupants.

FONDEN acted primarily in the Yucatan peninsula due to its high exposure to hurricanes. Suárez [26] states that hurricanes have been the climatic hazard that has caused the most damage in the peninsula since records began and the occurrence of tropical storms

corresponds to on average 10 per year with diverse intensities, including those that turn into hurricanes. The situation in the peninsula is expected to become more critical over the coming years as a consequence of climate change, which is known to increase the occurrence of extreme weather events [2].

Beneficiaries of FONDEN houses are low-income families. In the economic context, according to Castro del Rio (2009) [27], the lowest-income population in Yucatan is located in the rural zones, making them the most vulnerable to disasters. To contextualise the extent of poverty there, the National Council for the Evaluation of Social Development Policy (CONEVAL) has adopted a multidimensional approach to defining poverty. On the first dimension, poverty is measured according to different household income thresholds. In Yucatan, 46.3% of the population earns below the poverty line and is classified as living in poverty, and 12.5% of the population had earnings below the extreme poverty line. On the second dimension, poverty is measured according to the extent of deprivation of capabilities. For example, 19% of the population struggles with food access, 14.1% do not have access to health services, and 55.3% do not have access to welfare services. Finally, in terms of housing, at least 13.6% of the population lacks housing, and 38% do not have access to essential housing services (running water, drainage, electricity, and gas) ([28] p. 46, [29] p. 1).

Despite the need for housing, research has suggested that FONDEN houses have been abandoned due to low satisfaction rates among occupants [30], thus preventing FONDEN houses from remaining functional throughout their expected lifespan (>25 years) [26]. In this paper, the authors investigated the satisfaction levels of occupants of post-disaster FONDEN houses in Yucatan, Mexico, and identify key aspects that, according to the literature, have the most significant impact on house acceptance among the beneficiaries from the perspective of thermal comfort. The study aimed to identify aspects of the design that could be improved in future post-disaster housing to achieve higher levels of satisfaction among occupants.

The objective was to understand occupants' perspective and find out if the post-disaster social housing was fit for purpose and if thermal comfort has a role in the phenomena of the abandonment and repurposing of FONDEN houses and occupants going back to vernacular houses. The originality of this research is found in the thermal comfort evaluation of post-disaster social housing compared with the vernacular houses, monitored simultaneously in identical environmental contexts and also the correlation of them with the habitability and satisfaction rates among occupants. Despite there being a number of POEs in Mexico, their focus is on working environments, addressing different issues. The methodological approach taken in this study was POE, whereby the house occupants provided feedback and rated technical and functional aspects of the house. The POE included three different sources of data to support the investigation: a questionnaire, observation through photographic documentation, and environmental monitoring that will be further explained in the methods section.

### 1.1. Post-Disaster Social Housing

As previously forementioned, most of the post-disaster frameworks available were created by developed countries to recover from the economic loss expected after climatological hazards, aiming for an improvement in resilience during the recovery phase, and mostly relying on private insurance agencies.

In developing countries, on the other hand, during the recovery phase after a disaster, reconstruction commonly relies on governments and NGOs to fund and provide practical support for reconstruction, specifically for low-income housing [3]. This process can be conducted from two different approaches: the donor-driven approach and the user-driven approach [20].

The donor-driven approach delivers structurally sound homes but tends to overlook the social and cultural needs of the beneficiaries, often delivering standardised houses with repetitive patterns and typologies that are correlated with low satisfaction rates and high

rate of abandonment and repurposing. On the other hand, the user-driven approach is led by the beneficiaries, taking their cultural and social needs as a priority in the design, achieving a high rate of satisfaction [31]. However, if the construction process is not supervised and the builders are not trained, there is a risk of delivering poor structural quality, compromising occupants' safety [32].

Literature has shown that the most sustainable approach is generated by mixing the donor-driven and user-driven approaches by integrating the occupants into the decision-making during the design process, then training the beneficiaries to build their houses and provide specialised technical supervision [33]. This way, building knowledge stays in the community, comfort and family needs are met, and maintenance is sustained over the long term.

Eight indicators are presented by Carrasco et al. (2016) [14] in an analysis of satisfaction with permanent housing.

1. Location
2. Size of the plot
3. Size of the house
4. Construction quality
5. Strength of the house
6. Thermal comfort
7. Acoustic comfort
8. Functionality

In Mexico, in the occurrence of a disaster, the response phase starts with the state government declaring the zone as a "disaster zone" to the national authorities. After this, the federal organisation initiates the national procedures of Plan DN-III. The DN-III is the national post-disaster action plan, and it is managed by the Federal government, which through the Army and the air force reaches the emergency areas [34]. They act on five key points, which are: (i) searching and rescuing, (ii) evacuating communities, (iii) managing refuges, (iv) making recommendations to the population, and (v) taking care of the security and surveillance of the zone. However, this action plan has not been updated since 1966, and it takes an undifferentiated approach across the country, despite the evident differences in geographical, climatic, and cultural contexts [34].

The Mexican government, in 1996, began assigning 0.4% of the Federal budget (around 800 million USD) to fund prevention, mitigation and reconstruction related to disasters through a fund called FONDEN [24]. It covered damages that exceeded the response capacity of local and state authorities ([35] p. 16). They were in charge of designating funds to reconstruct public infrastructure and low-income housing once local governments declare an emergency state [3].

In the case of Yucatan, according to the post-disaster recovery plan published by the Yucatan State government [36], the Department of Public Security is responsible for managing the recovery funds. These funds are earmarked for (i) rehabilitation of public infrastructure, (ii) low-income housing reconstruction, and (iii) disaster risk reduction.

However, the recently (2022) published public guidelines for the reconstruction process after disasters keep a donor-driven approach that oversees cultural and specific needs of the local communities [37]. Once the local authorities receive the allocated fund, they distribute them to cover all the pertinent recovery costs, including bidding for local builders to carry out the construction work following the FONDEN house architectonic plans; thus, it is essential to assess these houses' suitability.

### 1.2. Housing in Yucatan

In the Yucatan peninsula, there are three main construction systems: (1) the vernacular, (2) the Colonial, and (3) the modern system.

### 1.2.1. Vernacular Houses

The vernacular house (Figure 1) in the geographical area considered in this study is composed of a single room with a size of 9 m by 5 m, typically with an apsidal floor plan and a thatched roof of 4 m height, the roof's lowest point being around 1.7 m—([26] p. 17).

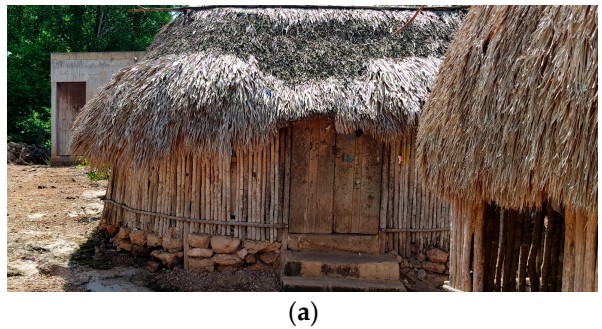
(**a**)

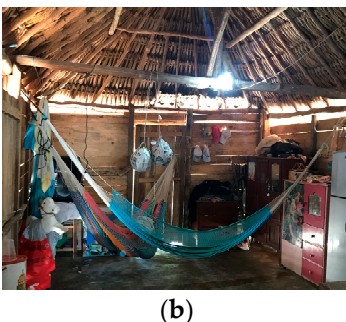
(**b**)

**Figure 1.** (**a**) Exterior view of a vernacular house, (**b**) interior view of a vernacular house. Source: author.

Vernacular houses typically have two double doors allowing an opening of 1 m on each side, one in front of each other, and they are 1.7 m high and have no windows. However, natural ventilation is enhanced through the structure, having gaps between the vertical components of the external wall and through the thatched roof. The house is constructed with locally sourced natural materials, including stones, earth, leaves, and tree branches [38].

The $U$-value of the thatched roof was calculated according to Equation (1) ([39] p. 7), where $U$ is the thermal transmittance in (W/m$^2$·K) and $R_{tot}$ is the total thermal resistance (m$^2$·K/W), determined via Equation (2) according to ISO-6946 (2017) ([39] p. 6), where $R$ is the thermal resistance (m$^2$·K/W), $d$ is the thickness of the material (m), and $\lambda$ is the thermal conductivity (W/(m·K)).

$$U = \frac{1}{R_{tot}} \tag{1}$$

$$R = d\lambda \tag{2}$$

For this case, the thickness of the material was 0.3 m and the thermal conductivity was 0.07 W/(m·K) according to CIBSE (2015) ([40] pp. 3–40).

$$R = \frac{0.3 \text{ m}}{0.07 \text{ W/m·K}} = 4.28 \text{ m}^2\text{K/W}$$

$$U = \frac{1}{4.28 \text{ m}^2 \text{ K/W}} = 0.23 \text{ W/m}^2\text{·K}$$

For the walls, which have discontinuous geometry, we used the $U$-Value of 0.71 W/m$^2$·K suggested by CIBSE (2017) for timber walls with 15% of thermal bridges caused by gaps resulting in air circulation between the interior and exterior parts of the building that are between 89 and 25 mm thickness ([39] pp. 3–12, [40] pp. 3–12).

### 1.2.2. Colonial Houses

Colonial housing was built in the zone from the XVI century to the early 20th century (Figure 2). The constructive system includes 40 to 90 cm thick limestone walls and flat concrete roofs over wooden beams cast in situ [41]. It features high ceilings ranging from over 4 m to 7 m in height, rectangular and extensive floorplans, and casement windows and doors that allow cross ventilation [41]. The $U$-value of a wall of a colonial house could vary from 1.49 W/m$^2$·K to 0.66 W/m$^2$·K, depending on the wall thickness [42].

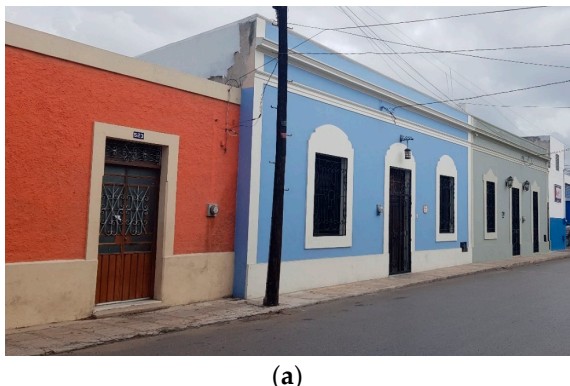 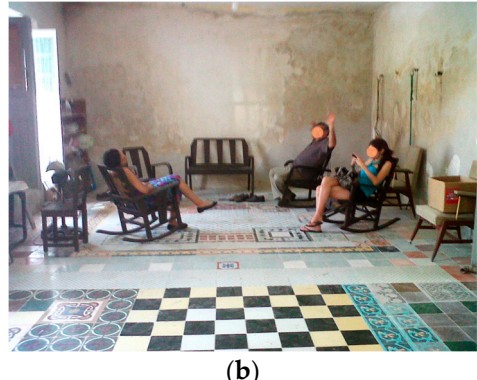

(**a**)                                                   (**b**)

**Figure 2.** (**a**) Exterior view of a colonial house, (**b**) interior view of a colonial house. Source: author.

### 1.2.3. Modern Houses

During the XX century, with industrial development, the construction system changed into a more cost-efficient and quick-to-build system based on hollow concrete blocks and industrialised modular T beams and hollow concrete filler blocks as roofing systems [43] as exemplified in Figure 3a,b. These building fabrics corresponded to a thermal transmissivity of 2.84 W/m$^2$·K for the walls and roofs [44,45] p. 280.

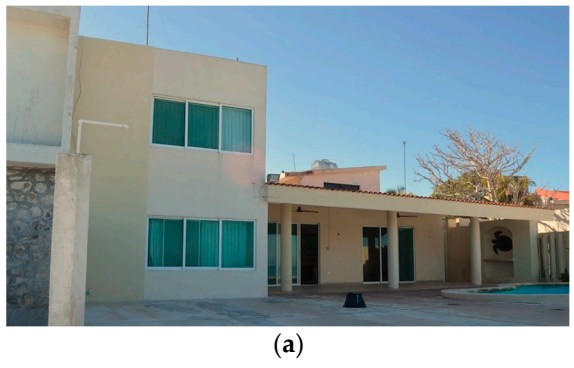 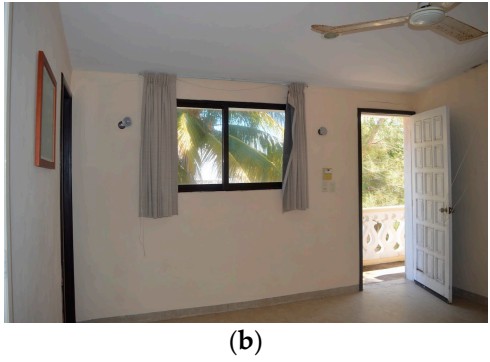

(**a**)                                                   (**b**)

**Figure 3.** (**a**) Exterior view of a modern house, (**b**) m interior view of a modern house. Source: author.

## 2. Materials and Methods

### 2.1. Thermal Comfort

Thermal comfort is defined as the "condition of mind that expresses satisfaction with the thermal environment and is assessed by subjective evaluation" ([46] p. 3). Comfort is an uncountable feature to consider in a house cost-effectiveness assessment [47] and it is relatively more difficult to demonstrate in comparison to energy consumption. It impacts occupants' health, mood, welfare, and productivity. Literature suggests that it represents an essential factor to consider when building sustainable [48].

The adaptive thermal comfort model considers the summation of several factors including the physics of the body, heat balance, climate conditions, social context, economic context, expectations, and preferences [46]. Thus, operative temperature calculation requires the following factors: convective heat transfer coefficient, air temperature, radiative heat transfer coefficient, and floor temperature [49]. However, this environmental study provided exclusively dry bulb temperatures and relative humidity data. Therefore, ASHRAE'S acceptability limits were used to analyse the environmental data collected.

The case studies were occupant-controlled, naturally conditioned spaces where the prevailing mean outdoor temperature ($t_{pma(out)}$) remained between 10 °C and 33.5 °C, the occupants had metabolic rates ranging from 1.0 to 1.5 met, and they were free to adapt their clothing insulation as desired. For that reason the following equations were considered

to analyse the acceptable indoor operative temperatures $t_o$ using the 80% of acceptability limits according to Equations (3) and (4) based on ASHRAE (2020) [46,50] p. 20–21.

$$\text{Upper 80\% acceptable limit } (^{\circ}\text{C}) = 0.31\, t_{pma(out)} + 21.3 \tag{3}$$

$$\text{Lower 80\% acceptable limit } (^{\circ}\text{C}) = 0.31\, t_{pma(out)} + 14.3 \tag{4}$$

Prevailing mean outdoor temperature $t_{pma(out)}$ considered the meteorological monthly means of each of the studied months according to Figure 4.

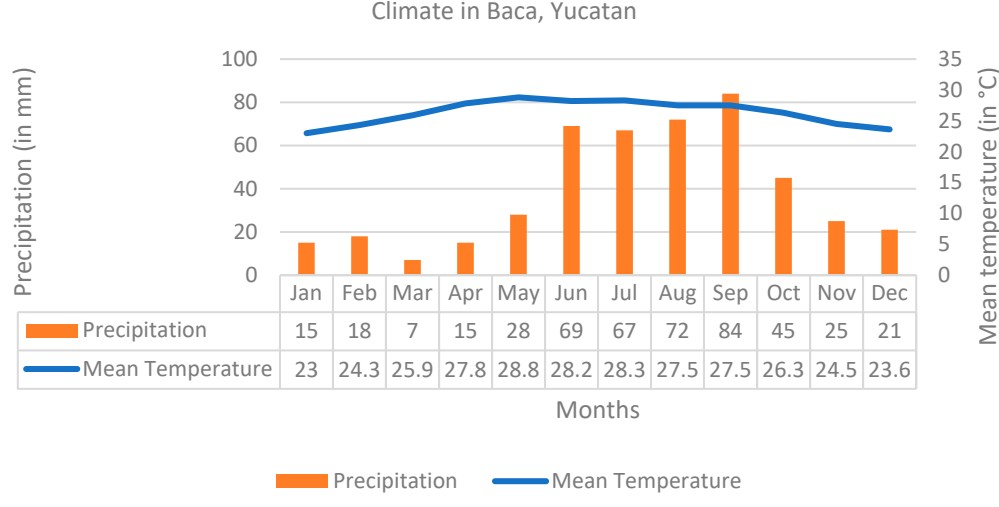

**Figure 4.** Local climate in Baca, Yucatan, using weather data from [51].

The adaptive thermal comfort model is mainly used in naturally ventilated buildings, where occupants play an essential role while thermally adapting to the environment; they tend to take action whenever they feel thermally uncomfortable. Those actions are called adaptations, and they can be made in the built environment or in the individual who feels uncomfortable. There are three personal thermal adaptations which are presented as follows: (i) behavioural, which refers to actions occupants take to improve their comfort in the room, (ii) psychological, based on the subjective perception of the occupant who tends to accept certain conditions due to a perception of control, thermal expectations, or forgiveness, and (iii) physiological acclimation, which is the adaptation made by the human body as a result of the effects of the environment ([52] pp. 14–19). This study analysed, through a POE, which dynamic or fixed comfort adaptations were made by occupants in the FONDEN houses after 17 years of use.

A post-occupancy evaluation (POE) was carried out on occupants of the FONDEN houses, with particular attention paid to thermal comfort. The POE sought to understand how users rated their original FONDEN design and also gain insight into the adaptations made by the users to meet their needs.

### 2.2. Case Study

The case studied was a donor-driven post-disaster housing reconstruction programme held in Mexico after Hurricane Isidore in 2002. During the reconstruction phase, a mass construction of 34 thousand houses was built following the local modern housing construction system [26]. The new houses were built in the land plot of the victims whose houses were severely damaged by the hurricane.

The typical FONDEN house has a rectangular geometry with a flat roof, its design consists of an open space and bathroom area with a total of 24 m² and a floor–ceiling height of 2.4 m (Figure 5). The building fabric includes ground flooring made of a 50 mm concrete layer, hollow concrete block walls that provide a *U*-value of 1.88 W/m²·K, and a T beam roofing made of hollow concrete filler blocks that provide a *U*-value of 1.06 W/m²·K on

the roof [44]. The building fabric is complemented with two iron sheet access doors, and three top-hung or sliding windows with a single 3 mm clear glass pane corresponding to a *U*-value of 5.90 W/m²·K.

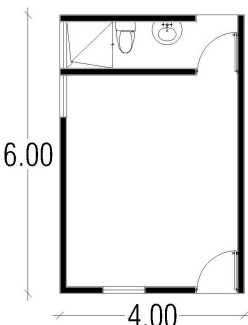

**Figure 5.** Floorplan based on the FONDEN official document indicating the general measurements of 4 by 6 m [24] and confirmed during the fieldwork.

FONDEN houses were occupied by low-income households. Figure 6a illustrates a typical FONDEN house as built and its aspect after 17 years of use. Figure 6b displays an interior view of a typical FONDEN house as built, with the typical living conditions and cultural adaptations held inside.

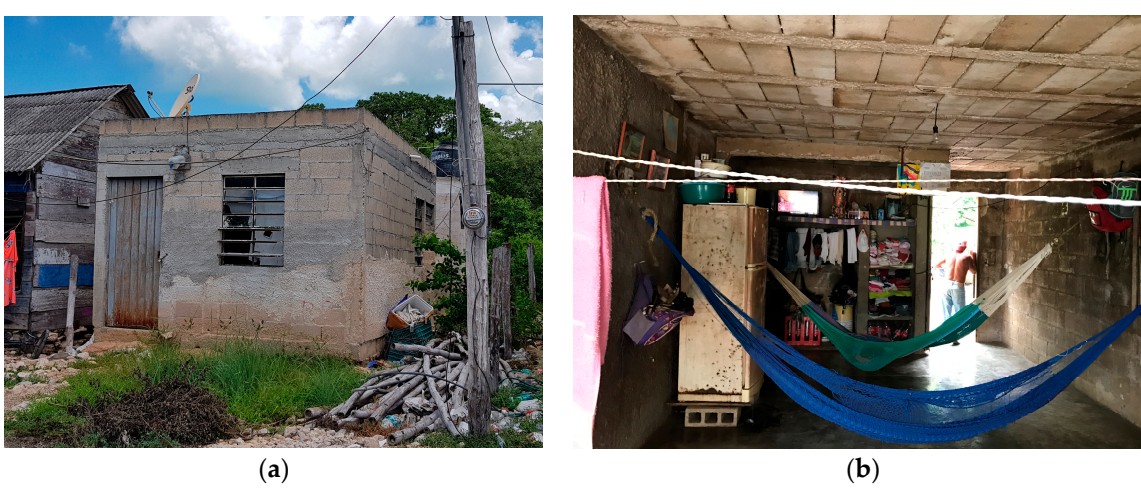

| (**a**) | (**b**) |

**Figure 6.** (**a**) Exterior view of a FONDEN house, (**b**) interior view of a FONDEN house. Source: author.

The study was conducted with participants from eight different municipalities in Yucatan (Mocochá, Baca, Calotmul, Espita, Sucilá, Panabá, San Felipe, and Merida). The studied municipalities were selected according to their diverse poverty level as reported by the National Council for the Evaluation of Social Development Policy in 2015 [25] from more than 40 communities that benefited from the aid provided after Hurricane Isidore in 2002. This allowed the analysis of a diverse sample from the lowest-income to the highest-income municipalities.

The POE included (i) a 25-question paper-based survey distributed to 103 FONDEN house occupants who were 18 years old or over, plus 13 abandoned houses that were not considered for the questionnaire given the absence of occupants, (ii) observation through photographic documentation, where a total of 83 participants agreed to welcome the researcher into their houses to take pictures and collect data from their lifestyles, and (iii) environmental monitoring in 2 dwellings from the same participant, a FONDEN house and a vernacular house for comparison purposes.

It is worth mentioning that the key differences between the vernacular houses and the FONDEN houses are related to their building fabrics, shapes, roofs, heights, and

fenestrations. The only similarities were found to be the single-room flexible floorplan and the orientation of the houses.

### 2.2.1. Questionnaire

Through the questionnaire (Appendix A), 103 participants shared their experience and satisfaction rates regarding FONDEN house characteristics such as (i) shape, (ii) materials, (iii) lighting, and (iv) thermal comfort. In addition, the questionnaire revealed the house acceptability in their cultural, economic, and social contexts.

The 25-question paper-based survey was designed to investigate the factors that can impact occupants' satisfaction levels with their houses. Participants were asked to answer about general demographics, type of tenure, and improvements in the buildings. A complete section was dedicated to rating, from 0 (highly dissatisfied) to 5 (highly satisfied), their satisfaction levels with their indoor environment, perception of safety, location, thermal comfort during cold and hot weather, daylighting levels, noise insulation, and materials. This data provided insights into key aspects to be improved in the FONDEN houses.

### 2.2.2. Observation through Photographic Documentation

Following the questionnaire, participants were invited to continue contributing to the study. This step included a visit to their house for photographic documentation. The photographic records aimed to provide insights into occupants' key strategies for adapting their home to achieve higher thermal comfort levels; the strategies' repetition was assessed through statistical analysis to identify the most common adaptive actions for comfort. Only 73 out of 83 properties visited were used for housing purposes and were considered for further comfort analysis.

### 2.2.3. Environmental Monitoring

Once the questionnaire and observation stages were covered, a single participant was invited to join the third stage of the investigation, for which the requirements were to have a property with both a FONDEN house and a vernacular house to simultaneously record thermal conditions in the same location. The selected property was located in Baca, Yucatan, and the thermal monitoring was scheduled for a representative week in the months of September and December. These were selected due to the fact that they are critical periods in terms of high air temperatures associated with high levels of humidity and low air temperatures combined with low levels of humidity, respectively.

Due to the intention of monitoring a FONDEN house and a vernacular one at the same location, the investigation was limited to these two housing units. The authors acknowledge the limitation of the study and support that more houses should be monitored in the future for a more representative sample.

The environmental monitoring collected dry bulb temperature and relative humidity data every 1 min. A total of four data loggers (tiny tags) were installed during each monitoring period and set to start and finish their recording simultaneously. Data loggers were calibrated and have a reading range from $-25\,°C$ to $+85\,°C/$ 0 to 95%RH of temperature and relative humidity, respectively, delivering an accuracy of $\pm0.6\,°C$ and $\pm3$%RH ([53] p. 2). The mean radiant temperature was not measured.

As described in Figure 7 through the green dots, the first and second devices were installed in the interior of the vernacular house, one positioned at the top of the ceiling (4 m) and the other at the height of people's activities (1.80 m). The third device was placed in the FONDEN house, also located at the same height of 1.80 m. Finally, a fourth data logger was installed at the same height of 1.80 m in an open-air area between the two houses to record the exterior conditions. This external device was protected from solar radiation through the overhanging thatched roof (Figure 7).

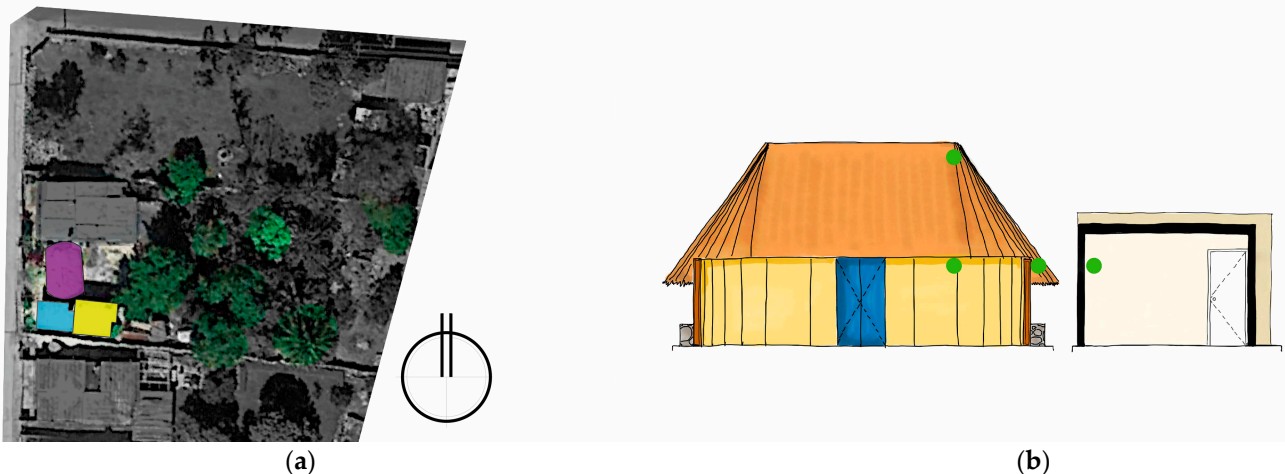

(**a**)　　　　　　　　　　　　　　　　　　　　　　　　　　(**b**)

**Figure 7.** (**a**) Location of FONDEN house and vernacular house in the land plot, (**b**) location of data loggers inside and outside the FONDEN and vernacular houses represented by green dots.

The authors acknowledge that the position of the dataloggers were not within the recommended height of 1.1 m above the floor, which accounts for the head for seated occupants ([46] p. 4). However, the study was conducted during the time of use of these indoor spaces and monitoring had to prevent disturbance to the occupant's daily activities, as well as to prevent the reach of children whose hands temperature could damage the accuracy of data. The adopted height is similar to the values of 1.7 m for standing occupants ([46] p. 4).

The FONDEN house (blue in Figure 7a) was built next to the vernacular house (purple in Figure 7a), 60 cm from each other; both facades were facing west, with the FONDEN house having one window placed on the west façade and the vernacular house one double door. Data collected were then analysed in Microsoft Excel to calculate the percentage of time dry bulb temperatures were below, within, and above acceptability thresholds according to Table 1.

**Table 1.** Prevailing mean outdoor temperature $\left(t_{pma(out)}\right)$, lower acceptability limit ($t_{min}$) and upper acceptability limit ($t_{max}$) in Baca, Yucatan, calculated according to Equations (3) and (4) for a 80% of acceptability.

|  | $t_{pma(out)}$ | $t_{min}$ | $t_{max}$ |
|---|---|---|---|
| September | 27.5 °C | 22.8 °C | 29.8 °C |
| December | 23.6 °C | 21.6 °C | 28.6 °C |

## 3. Results and Discussion

Data were statistically analysed through Microsoft Excel to make an effective cross-comparison between the different sections and understand the correlations between the results.

### 3.1. Questionnaire

The analysis from the questionnaire suggested that only 11% of the occupants abandoned their houses, while 83% of the occupants were still using their FONDEN houses for habitational purposes and 6% of the participants had repurposed their homes. The other purposes given to the homes were kitchen, storage room, temporarily rented for business, or own workplace.

Regarding structural reliability, 53% of the participants said they considered their house structurally safe enough to stay in during a hurricane, while 47% claimed they would evacuate their homes during a hurricane.

In terms of the physical building, 60% of the participants reported they made changes or improvements to their homes. The most common improvement (56%) was related to the protection of the building envelope from weathering by adding render on the roof and walls, which served a primary purpose of protection against rain and humidity by (i) avoiding leaking while raining and (ii) diminishing general relative humidity levels coming through the building envelope. Participants also revealed improvements they had planned but had not had the opportunity to implement them. For example, 33% wanted at least one extra room for the kitchen or bedroom, suggesting that the typology did not offer enough space to develop daily activities. In addition, 26% of the participants wanted to improve their toilets/drainage, 25% mentioned a relocation or insertion of a door, 15% mentioned that wanted to add ceramic flooring, and 11.4% wanted an improvement in their windows, including relocation or reducing glazing. These results indicate the need for adequate floor space and quality building fabrics to perform their daily activities safely and comfortably.

Satisfaction rates (Figure 8) showed that overall participant perception was generally positive due to the excellent timing of the housing provision, just after losing their original homes. Furthermore, on the one hand, it was noticeable that the most positive answers were regarding location and ability to perform their social and domestic activities; this might be related to the flexibility that the main room allows in complement with the cultural adaptations. For instance, the ability to roll their bedding systems up during the day and use the space for other purposes is similar to the flexibility of the vernacular homes. On the other hand, questions with mostly negative responses were related to thermal comfort in warm weather, with 74%, and cold weather, with 43% of occupants dissatisfied. This might be related to a lack of adaptive controls to adjust the house to their needs and the generally inadequate thermal comfort delivered by the houses.

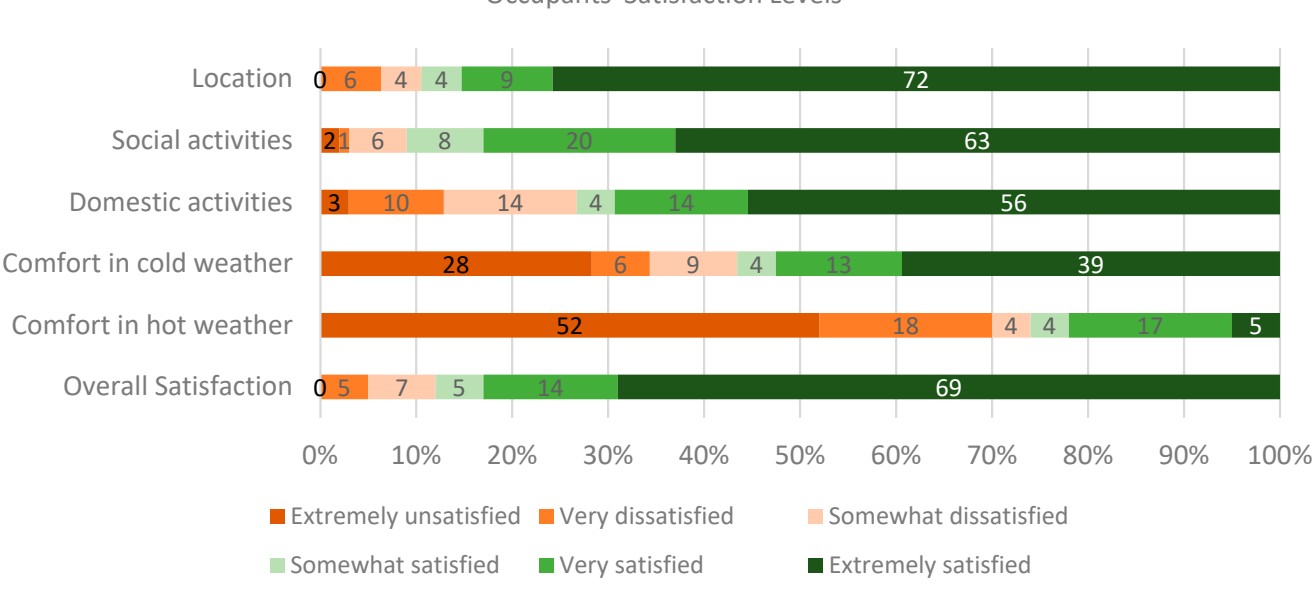

**Figure 8.** Occupants' satisfaction rates.

It was notable the importance participants gave to the finishing of their homes when asked "what improvements they have made to their homes?" (Figure 9), resulting in an insight into their perception of potential adaptive controls that could be integrated into the homes to make them more suitable for their lifestyle.

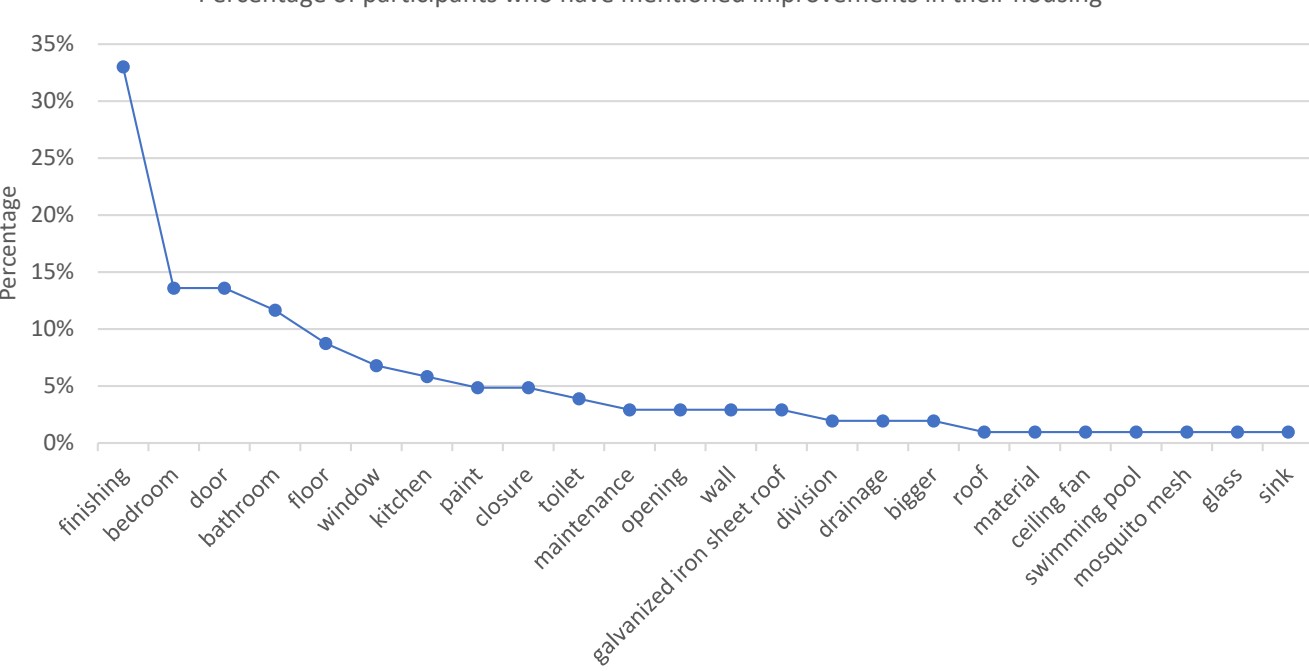

**Figure 9.** Percentage of mentioned improvements implemented in the participants' houses (each house could have more than one improvement).

*3.2. Observation*

The observation was carried out in 83 houses and further investigated key aspects of use and occupancy indicated by occupants' responses to the questionnaire. The majority of the occupied houses were found to be in a bad state of repair just 17 years after their construction, even though the motivation of FONDEN houses was to deliver durable and resilient housing units, and the expected lifespan of a concrete-based house in a tropical zone should be between 25 and 30 years ([26] p. 163). The most common improvements made by the occupants were the integration of extra layers to the walls, such as rendering or plaster, as Figure 10 exemplifies.

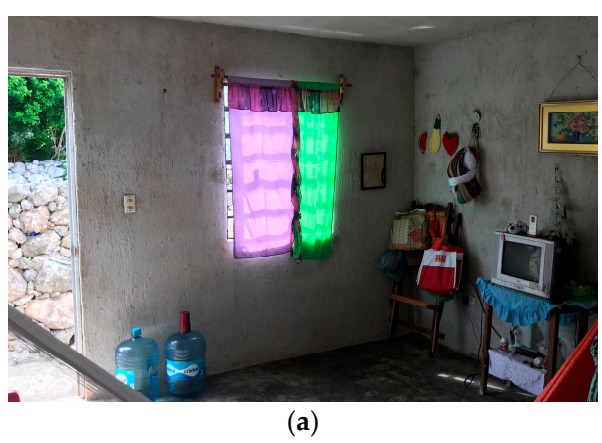

(**a**)             (**b**)

**Figure 10.** *Cont*.

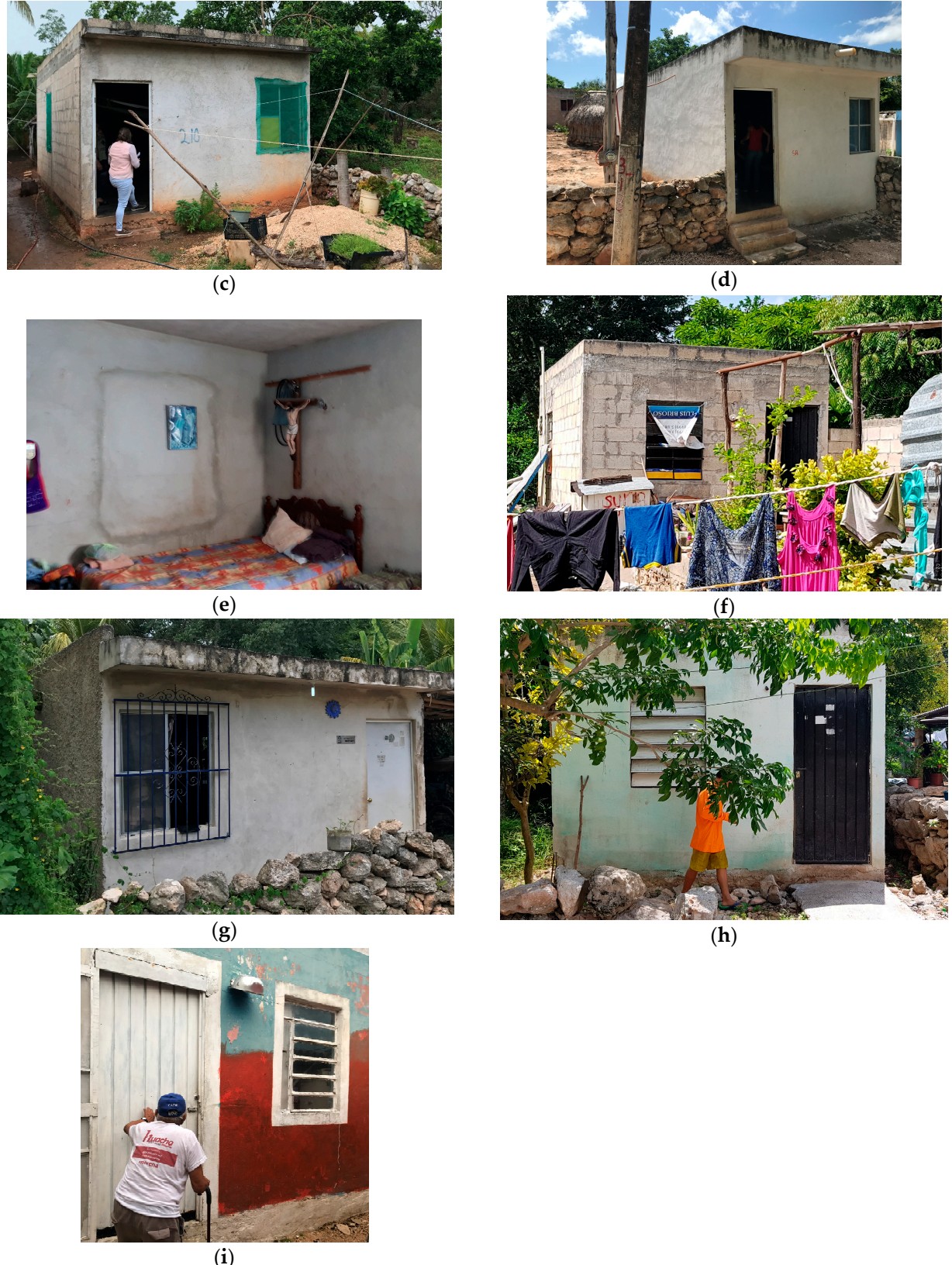

**Figure 10.** Example of improvements made by occupants to their FONDEN houses. (**a**) Interior curtains, (**b**) attached lightweight structure, (**c**) mosquito mesh, (**d**) canopy, (**e**) cancelling windows, (**f**) exterior curtain-like shading, (**g**) safety protection, (**h**) painting window glazing, (**i**) painting door white. Source: authors.

Figure 11 presents the figures of the presence of these strategies in the observed homes to adapt their house to achieve indoor thermal comfort.

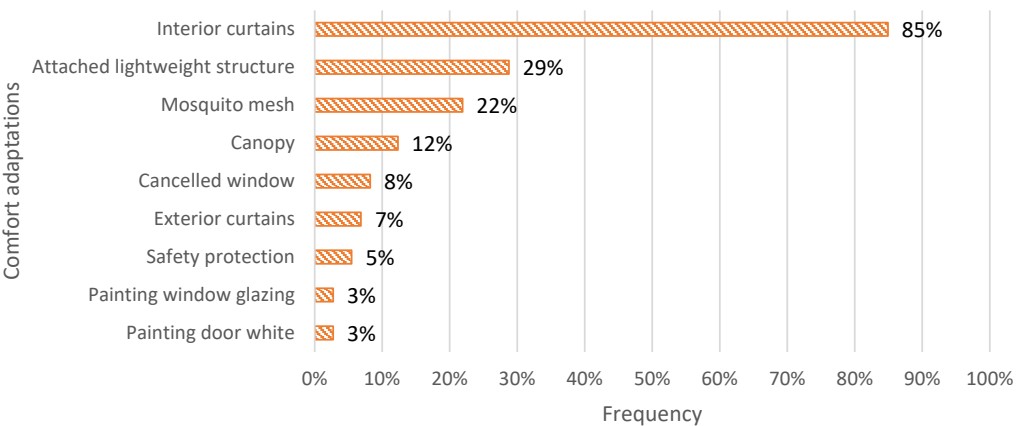

**Figure 11.** Percentage of building control systems in the surveyed homes.

In each of the houses visited during the photographic documentation, it was noticed that most of the occupants introduced electric appliances to improve their comfort levels (Figure 12). It should be mentioned that most of the observation work was carried out in July 2019. During the corresponding season (summer), the prevailing mean outdoor temperatures remained above 23 °C for nearly 46% ([54] p. 37) of the time, which suggests a risk of overheating and significant demand for cooling.

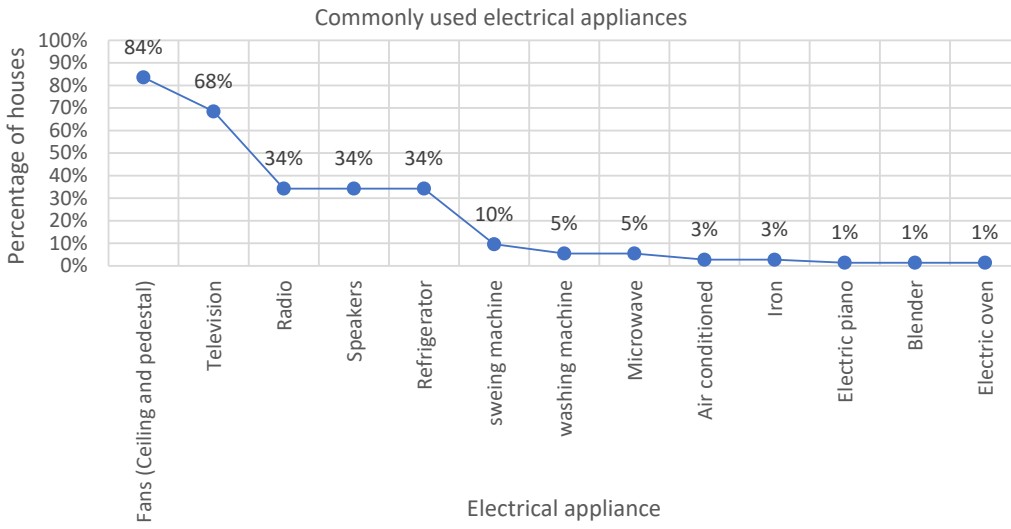

**Figure 12.** Percentage of the presence of common electrical appliances in documented homes.

Figure 12 illustrates the percentage of houses that accounted for at least one of the appliances listed. From observing the ownership of electrical appliances, it was possible to notice that the average number of electrical appliances per house was three. Surprisingly, fans were the most common device, and appeared in 84% of the houses surveyed, followed by televisions, presented in 68% of the houses, and radios, speakers, and refrigerators, which appeared in 34% of the houses.

In addition, air-conditioning units were found in two houses, regardless of their high purchase, installation, and running costs. This was unexpected, given that most of the beneficiaries' income corresponds to up to 5.5 USD a day [24]. In one of the cases, air conditioning was running while the household did not feature a functional toilet. This provides insight into participants' needs, placing thermal comfort on top of sanitary facilities, suggesting that comfort might be one of the key aspects to be improved in the FONDEN houses.

*3.3. Monitoring*

As seen in Figure 13, the analysis of the monitored data indicated that the FONDEN house was uncomfortable for 67% of the time, when outdoor conditions were warm and humid (in September), whilst the vernacular house was uncomfortable for only 29% of the time under the same outdoor conditions, revealing the better suitability of the vernacular house to warmer conditions. This seems to be due to the high ventilation rates and specific thermal properties of the building fabric in the vernacular building.

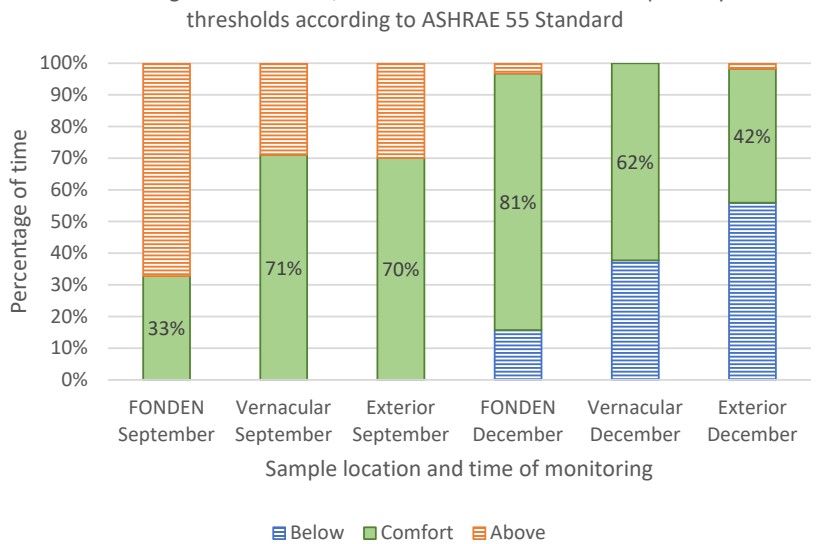

**Figure 13.** Percentage of time below, within and above the 80% acceptability thresholds according to ASHRAE 55 Standard.

On the other hand, in December, when temperatures were lower, FONDEN houses were found to be within thermal comfort for 81% of the time, while vernacular houses, were comfortable for 62% of the time (Figure 13). This revealed that the FONDEN houses are more suitable for colder environments, while the conditions in the zone are generally warmer. This is because the vernacular house allows more ventilation due to the characteristic of the wall, while the FONDEN house has a more airtight envelope comparatively with limited operable doors and windows, which reduces the airflow and, consequently, the heat transfer through the envelope.

As seen in the questionnaire, building openings are mostly kept closed during the night for safety reasons. During warmer months, the concrete has a higher thermal mass that releases the heat at night, creating a warmer environment than outdoors. On the other hand, during the cold season, this effect causes a more comfortable environment because exterior air temperatures are lower than the comfort zone and the heat released by the concrete enhances comfort in FONDEN houses. For that reason, ventilation and its optimisation are key factors in the building to enhance comfort in warmer periods.

Despite the fact that the PMV model was conceived for mechanically conditioned spaces where the adaptation phenomena are limited [55,56], a further analysis based upon the well-known ASHRAE thermal comfort chart was carried out. The psychrometric charts (Figure 14) illustrate monitored data of dry bulb temperature and humidity data plotted against the thermal comfort threshold, shown in green. FONDEN and vernacular houses were comparatively analysed for the two periods monitored, which were September and December. The threshold considered for this analysis was chosen according to Table 1, assuming the typical clothing insulation for summer equivalent to 0.5 clo for people who are engaged in near-sedentary physical activity [46].

|  | FONDEN house | Vernacular house |
|---|---|---|
| September | | |
| December | | |

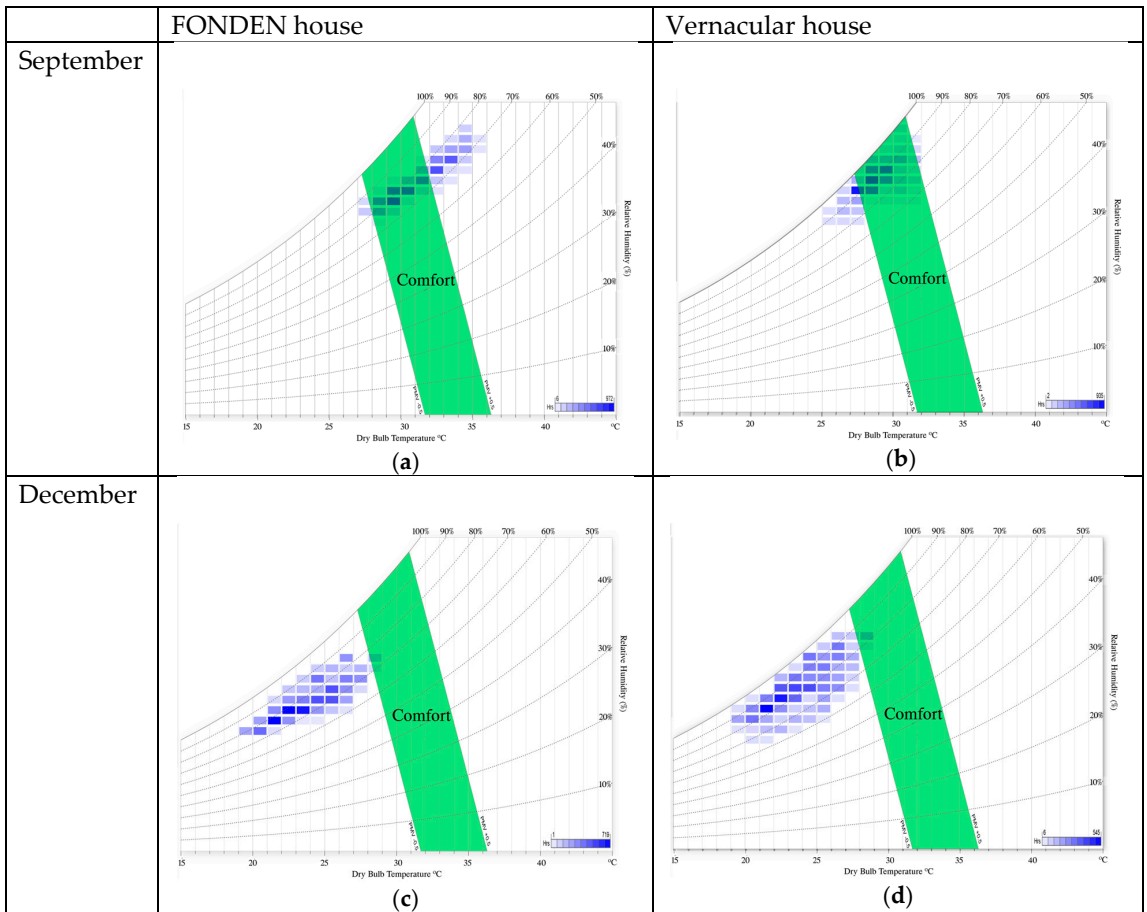

**Figure 14.** (**a**) Psychrometric chart of environmental conditions recorded in FONDEN house in September, (**b**) psychrometric chart of environmental conditions recorded in vernacular house in September, (**c**) psychrometric chart of environmental conditions recorded in FONDEN house in December, and (**d**) psychrometric chart of environmental conditions recorded in vernacular house in December. Assumptions: air velocity value <0.1 m/s and mean radiant temperature equal to air temperature (uniform conditions).

Considering this, most of the recorded conditions from Figure 14 remained out of the thermal comfort zone, above thermal comfort in September and below thermal comfort in December, both with high levels of humidity (above 80%). The humidity levels were higher in the vernacular houses on both occasions; a combination with lower dry bulb temperatures in September delivered more comfort than in the FONDEN house. However, conditions were less comfortable during winter, since dry bulb temperatures were lower than in the FONDEN house.

## 4. Conclusions

The donor-driven approach in post-disaster social housing programme in Mexico has demonstrated to present some drawbacks for the end users, including thermal discomfort leading to repurposing and abandonment. This research has suggested through a POE that FONDEN houses in Mexico, which have received annual investment of USD 800 million, could be improved to deliver more comfortable, resilient and sustainable housing for low-income households.

This study evaluated the post-disaster FONDEN houses and used three different research methods. These were: questionnaire, observation through photographic documentation, and environmental monitoring. The research aimed to assess occupants' satisfaction levels and thermal comfort delivered in these dwellings.

Cross-comparing data from the study, the results corroborate that poor thermal comfort in the houses was an issue found in the typology investigated. Even though the results from the observations and questionnaires indicated that most of the beneficiaries were still occupying the houses (103 out of the 116 samples), 97 out of 103 surveyed participants (94% of the observed houses) indicated that the purpose of their house was still habitation. However, 52% of the participants were extremely dissatisfied with their houses.

In addition, data from observation through photographic documentation in the houses visited suggested that 61 households out of 73 samples (84% of the households) integrated an air movement devices, being fans (including the pedestal and ceiling types) the most common electrical appliance within the households. Data from monitoring revealed a predicted thermal discomfort for 92% of the time during September, when conditions prevailed above thermal comfort.

Poor thermal comfort was consistent in the houses throughout this study; according to the climate chart (Figure 4), worse comfort scenarios might be expected during warmer months (April to August).

This research highlighted the importance of thermal comfort as a design parameter in post-disaster social housing in developing countries. Within the findings, it was noticed that thermal discomfort was one of the main causes of dissatisfaction among occupants, suggesting a need for further investigation on how to improve thermal comfort in these houses.

Some limitations must be acknowledged, since the environmental devices for monitoring were placed at different heights from those typically adopted, as well the lack of the mean radiant temperature and air velocity readings. Moreover, for an accurate calculation of the prevailing mean outdoor temperature, it was necessary to account for the daily mean temperature of the previous (7 to 30) subsequent previous days. However, as a consequence of the lack of this information, meteorological monthly mean temperatures were considered for the calculation of the prevailing mean outdoor air temperature $\left(t_{pma(out)}\right)$, following the alternatives suggested by ASHRAE (2020) [46].

Further research is needed, including a much larger sample to make informed decisions about the optimisation of these houses. Additionally, further analysis is needed to quantify the impact of each building feature on thermal comfort and inform future post-disaster social housing programmes to ensure occupants' needs are met and suitability for the climate is addressed.

**Author Contributions:** Conceptualization, Y.A.-P.; methodology, Y.A.-P., L.R., R.T. and P.B.; software, Y.A.-P.; validation, Y.A.-P., L.R., R.T. and P.B.; formal analysis, Y.A.-P.; investigation, Y.A.-P.; resources, Y.A.-P.; data curation, Y.A.-P.; writing—original draft preparation, Y.A.-P.; writing—review and editing, Y.A.-P., L.R., R.T. and P.B.; visualization, Y.A.-P., L.R., R.T. and P.B.; supervision, L.R., R.T. and P.B.; project administration, Y.A.-P.; funding acquisition, Y.A.-P. All authors have read and agreed to the published version of the manuscript.

**Funding:** This research was funded by CONACyT and SENER, through the scholarship awarded to the first author for doctoral studies.

**Institutional Review Board Statement:** The study was conducted in accordance with the Ethics Committee of the Faculty of Engineering from the University of Nottingham approved on the 23 July 2019.

**Informed Consent Statement:** Informed consent was obtained from all subjects involved in the study.

**Data Availability Statement:** Not applicable.

**Acknowledgments:** The authors would like to thank the residents of the case studies, who opened their homes to the fieldwork as well as the volunteers, who helped with data collection. Also, the authors would like to acknowledge Architecture, Culture and Tectonics (ACT) and Buildings, Energy and Environment (BEE) research groups for supporting the publication.

**Conflicts of Interest:** The authors declare no conflict of interest.

## Nomenclature

| | |
|---|---|
| $\lambda$ | =Thermal conductivity, W/(m·K) |
| $d$ | =Thickness of material, m |
| $I_{cl}$ | =Clothing insulation, clo |
| $t$ | =Temperature, °C |
| $t_{max}$ | =Temperature for upper acceptability limit, °C |
| $t_{min}$ | =Temperature for lower acceptability limit, °C |
| $t_{pma(out)}$ | =Prevailing mean outdoor temperature, °C |
| $R_{tot}$ | =Total thermal resistance, m$^2$·K/W |
| $U$ | =Thermal transmittance, W/m$^2$·K |

## Appendix A

| 1.1 Age ______ | 1.2 Gender   Male ☐          Female ☐          Prefer not to say ☐ |
|---|---|

1.3 Locality ________________________

1.4 Did the Yucatan Government give you the FONDEN house?   Yes ☐   No ☐
If no, please specify if you rent it, bought it or was donated to you ________________

1.5 Did you make any change or improvement to the FONDEN house after receiving it?   Yes ☐   No ☐
Which? ________________
1.6 Do you use the FONDEN house for living?   Yes ☐   No ☐

1.7 Have you ever given any function to the FONDEN house, different from the current one?   Yes ☐   No ☐
Which? ________________

1.8 Were the FONDEN house's construction complete at the time of receiving it?   Yes ☐   No ☐

1.9 Would you stay in the FONDEN house during a hurricane?   Yes ☐   No ☐

1.10 If anything were missing, please tick in the boxes the parts that it was lacking or tick "other" and list them.
Windows ☐          Ceiling ☐
Doors ☐          Finishings ☐
Toilet ☐          Other ☐ ____________
Installations ☐

1. 11 Approximately how much time after Hurricane 'Isidore' have you been given the FONDEN house?
Less than 1 year          0 ☐     1 ☐     2 ☐     3 ☐     4 ☐     5 ☐          6 or more

1.12 Please rate your level of satisfaction of the FONDEN house as it was given to you
Completely dissatisfied          0 ☐     1 ☐     2 ☐     3 ☐     4 ☐     5 ☐          Completely satisfied

1.13 How did you find the FONDEN house to be adequate in relation to cold seasons
Completely inadequate          0 ☐     1 ☐     2 ☐     3 ☐     4 ☐     5 ☐          Completely adequate

1.14 How did you find the FONDEN house to be adequate in relation to hot seasons
Completely inadequate          0 ☐     1 ☐     2 ☐     3 ☐     4 ☐     5 ☐          Completely adequate

**Figure A1.** *Cont.*

**Figure A1.** Example of the paper-based survey applied to participants during the fieldwork.

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
