# Peer review of "Post-Occupancy Evaluation in Post-Disaster Social Housing in a Hot-Humid Climate Zone in Mexico"

_sustainability, doi:10.3390/su151813443_

Round 1
Reviewer 1 Report
Dear Authors,
here below you will find some general and specific comments about your manuscript.
Best regards.
Main criticalities
- The introduction is badly structured. The authors should describe and discuss the current state of the art (at the regional and worldwide stage), and then stress the peculiarities and the novelties of the current investigation. In addition, the reference seems to be not recent enough in some cases. Finally, all information about the case study and the methodology to assess comfort conditions (section 1.4) should be moved to the material and methods section.
- The environmental monitoring is poor due to the lack of measurement of the operative temperature.
- Some results are not supported by strong evidence. This is the case of Figure 13 data which is affected by the lack of operative temperature measurements.
- The conclusions should stress all the limitations of the monitoring campaign.
Some other specific comments
· Introduction
L60-L61. This sentence is not supported by references.
Equation 1 is not supported. Is maybe from ASHRAE 55 or EN 16798-1? Why only one equation if ASHRAE 55 reports upper and lower 80% acceptability limits? Please discuss and specify.
· Materials and methods
Indoor environmental monitoring is poor. This is why adaptive model requires the measurement of the operative temperature which requires the measurement of the mean radiant temperature (see http://dx.doi.org/10.1016/j.enbuild.2014.06.033). In addition, the height of indoor sensors is not conventional (see ISO 7726 and ASHRAE 55)
· Results and discussion
Figure 13 results are affected by the lack of measurement of the operative temperature.
About figure 14, is the sample worn by typical Icl values for summer (0,5 clo)?
· Symbols
The symbol for degree Celsius is °C (please modify everywhere within the paper). In addition a short nomenclature section is suggested.
· References
References from [1] to [4] are not recent enough.
The ASHRAE 55 standard has been recently updated.
Author Response
Dear reviewer,
The authors thank you for your insightful comments. They were very welcome and indeed helped to improve the paper.
Please find below a list of the main changes that were made to the paper:
- The introduction was re-structured, describing the current state-of-the-art post-disaster frameworks and stressing the originality of this investigation.
- When possible, references were updated to the last 5 years.
- Information on the case study and thermal comfort theory were moved to the materials and methods section.
- The limitations of the monitoring were discussed in the discussion section and acceptability formulas from ASHRAE were adopted for the comfort assessment criterion.
- L60-L61 was integrated with relevant references.
- Equations for acceptability thresholds were supported with reference to the latest version of ASHRAE 55
- Results in Figure 13 were updated according to the corrections made in Table 1.
- An acknowledgement of the limitations of the monitoring campaign was added to the conclusions including the unconventional datalogger height.
- It was clarified that the results considered a typical Icl value for summer (0,5 clo)
- ºC symbols were updated throughout the paper.
- A nomenclature section was added at the end of the paper.
Overall, the paper has been thoroughly revised to clarify the focus of the work, reflect that in the literature review, and revised the contents generally to ensure its contribution is concise, informative, and valuable to our research community.
Reviewer 2 Report
The article's subject fits very well with the thematic scope of the Sustainability journal. Therefore, it would be valuable to publish the text. However, the current submission version requires corrections and clarifications before the final acceptance.
1. The authors must precisely distinguish between dry bulb temperature and operating temperature in all referenced standards, measurements, calculations, etc. Please note that toc (equation 1) is the indoor comfort operative temperature. For air velocity below 0.2 m/s, an operative temperature equals the arithmetic average of dry air temperature and mean radiant temperature. Unluckily there is no discussion on mean radiant temperatures in analyzed buildings.
2. The text needs at least a broader discussion about the consequences of the location of loggers. For example, thermal comfort indoors is typically evaluated at 1,1 m above the floor. What were the typical vertical profiles of the dry bulb temperature in the analyzed buildings? Negligible? Can the values measured at 1,8 m be used to substitute values measured at 1,1 m? What was the impact of wall radiation on the logger readings?
3. Lines 248-251: "Thus, predicted thermal comfort thresholds vary according to the specific climate conditions and are determined by the equation (equation 1) to consider the psychological expectations of the occupants in the studied location." It seems like it would be more appropriate to rewrite it, e.g., "Thus, the predicted optimal indoor operative temperatures vary according to the specific climate conditions and are determined by equation 1 to consider the psychological expectations of the occupants in the studied location."
4. The authors should consider that de Dear and Brager's adaptive thermal comfort model includes optimum operative temperature and acceptability limits. (Please see De Dear, R., & Schiller Brager, G. (2001). The adaptive model of thermal comfort and energy conservation in the built environment. International journal of biometeorology, 45, 100-108.)
Comfort temperature (°C) = 0.31 (mean outdoor monthly air temperature) + 17.8
Upper 80% acceptable limit (°C) = 0.31 (outdoor air temperature) + 21.3
Upper 90% acceptable limit (°C) = 0.31 (outdoor air temperature) + 20.3
Lower 80% acceptable limit (°C) = 0.31 (outdoor air temperature) + 14.3
Lower 90% acceptable limit (°C) = 0.31 (outdoor air temperature) + 15.3
Which acceptability limits were adopted when presenting the results in Figure 13? ( I PMV I < 0.5, which corresponds to 90%? )
5. Table 2 contains some surprising data. Firstly, there is no clear explanation of what Tmin and Tmax are. Are they measured indoor temperatures or values calculated according to equation (1) based on actual outdoor measurements? What was averaging time of the readings? Moreover, the temperatures Toc, Tmin, and T max are identical for September and December. Why was the data reported differently, with one or two decimal places? Please remember that temperatures expressed in Celsius usually use the symbol t in contrast with temperatures presented in Kelvins, noted as T.
6. Figure 9 presents that only 1% of the residents indicated a fan as an improvement of the house they received. At the same time, Figure 12 shows that fans were the most typical electrical device used in 84% of the houses. The abstract presents inconsistent data that the percentage of dwellings with fans was 72%. A discussion is needed.
Author Response
Dear reviewer,
The authors thank you for your insightful comments. They were very welcome and indeed helped to improve the paper.
Please find below a list of the main changes that were made to the paper:
- Dry bulb temperature and operating temperature were distinguished throughout the paper.
- Equations for acceptability thresholds were supported with reference to the latest version of ASHRAE 55
- An acknowledgement of the limitations of the monitoring campaign was added to the conclusions including the unconventional datalogger height. Also, it was mentioned a similitude to ASHRAE suggested height for comfort when considering the individuals in a standing position.
- Lines 248-251 were edited, explaining the adoption of the adaptive model acceptability thresholds equations adopting the criterion for 90% of acceptability.
- Optimum operative temperature and acceptability limits were added to the description of the comfort assessment criterion.
- tmin, tmax, toc and tmean were explained in the body of the text and in the nomenclature section at the end of the paper. Also, the use of decimal places was homogenised, and the calculations were revised, resulting in a corrected Figure 12.
- Clarification about Figure 9 was made where only 1% of the participants had included a ceiling fan while Figure 12 shows that fans (pedestal and ceiling) were found in 84% of the houses.
Overall, the paper has been thoroughly revised to clarify the focus of the work, reflect that in the literature review, revised the contents generally to ensure its contribution is concise, informative, and valuable to our research community.
Round 2
Reviewer 1 Report
Dear Authors,
you authors must read carefully the AHSRAE standard 55 before revisiting your manuscript. This is why thermal comfort issues are superficially discussed (this is also for the applied methodologies). In addition, you did not answer punctually to my observations, so it was difficult for me to verify the effectiveness of their replies.
Here below you will find some other comments.
Best regards.
· In what manner U values for vernacular houses have been measured?
· Eq. 1 symbols are not consistent with ASHRAE 55. You must use symbols and related meanings. Tout is not the outdoor temperature or a mean temperature. It is the Running Average Outdoor Air Temperature. In what manner did you evaluate it? Consistently with ASHRAE 55? Figure 6 does not explain it. Average running temperature requires daily data not averaged monthly data.
· Figure 14 should be better discussed. This is why the PMV model cannot be applied in naturally ventilated buildings, but only in mechanically conditioned environments. Is this your situation? Please motivate. Did you measure air velocity to use the PMV? What about the metabolic rate value? Is the same for which figure 14 works? I suggest reading carefully this manuscript by DTU researchers (who formulated the PMV model): doi:10.3390/atmos11010049. This is also because you can apply the PMV model for indoor air temperature values below 30 °C.
· In the conclusions, another limitation is the air velocity lack of measurements and probably the way to asses the running average outdoor air temperature.
· The nomenclature section must be revised.
o Tmin and Tmax definitions must be consistent with definitions in ASHRAE 55: 2020. This is also for Tmean. It is the Running Average Outdoor Air Temperature (you must revise the manuscript).
o °C is not a symbol, it is a unit
o The "clothing level is wrong". You must use basic clothing insulation as indicated in ISO 7730, ASHRAE 55, and ISO 13731.
Author Response
"Please see the attachment."

Reviewer 2 Report
Thanks for thorough reacting to my comments.
Author Response
Thank you very much for your feedback.
Round 3
Reviewer 1 Report
Dear Authors,
here below you will find some comments about the application of PMV model in NV buildings.
Best regards
L577 – 583. The authors state: “Predicted mean vote (PMV) is mainly used to evaluate mechanically conditioned environments. In naturally ventilated buildings, such as this case, six environmental factors should have been analysed to accurately evaluate the PMV: dry bulb temperatures, relative humidity, metabolic rates, air speed and radiant temperature. However, this study considered exclusively dry bulb temperatures ad relative humidity, assuming an adaptive clothing insulation. Given this limitation, the PMV shown in Figure 14 would only be reliable for a uniform environment. In other words, an environment where mean radiant temperature and air temperature are similar.”
The authors are not accurate in the application of PMV in NV buildings. PMV index is conceived for mechanically conditioned environments and, for naturally ventilated (NV) buildings it works with some adjustment accounting for adaptive phenomena (see Toftum et al. and Attaianese et al. papers). In both cases (natural and mechanical conditioned spaces) six variables are required. Based on the reply by the authors (L577-583) and the argumentation as above, I suggest:
· inserting the sentence as follows before L 564 (by also integrating a couple of references):
Despite PMV model has been conceived for mechanically conditioned spaces where the adaptation phenomena are limited (Toftum et al., 2002, Attaianese et al., 2019), a further analysis based upon the well knows ASHRAE thermal comfort chart has been carried out.
o Fanger PO, Toftum J (2002) Extension of the PMV model to non-air-conditioned buildings in warm climates. Energy Build 34:533–536
o Attaianese E, d’Ambrosio Alfano FR, Palella BI (2019). An Ergonomic Approach of IEQ Assessment: a Case Study. Proceedings of the 20th Congress of the International Ergonomics Association (IEA 2018) Volume VIII 504–513. doi: 10.1007/978-3-319-96068-5_57.
· deleting lines 577-583;
· modifying the caption of the figure 14 as follows:
Figure 14. (a) Psychrometric chart of environmental conditions recorded in FONDEN house in September, (b) Psychrometric chart of environmental conditions recorded in Vernacular house in September, (c) Psychrometric chart of environmental conditions recorded in FONDEN house in December, and (d) Psychrometric chart of environmental conditions recorded in Vernacular house in December. Assumptions: air velocity value <0.1 m/s and mean radiant temperature equal to air temperature (uniform conditions).
